# Replication Stress Response Links RAD52 to Protecting Common Fragile Sites

**DOI:** 10.3390/cancers11101467

**Published:** 2019-09-29

**Authors:** Xiaohua Wu

**Affiliations:** Department of Molecular Medicine, The Scripps Research Institute, La Jolla, CA 92037, USA; xiaohwu@scripps.edu

**Keywords:** RAD52, FANCM, common fragile sites, replication stress, DNA double-strand breaks, homologous recombination, break-induced replication

## Abstract

Rad52 in yeast is a key player in homologous recombination (HR), but mammalian RAD52 is dispensable for HR as shown by the lack of a strong HR phenotype in RAD52-deficient cells and in *RAD52* knockout mice. RAD52 function in mammalian cells first emerged with the discovery of its important backup role to BRCA (breast cancer genes) in HR. Recent new evidence further demonstrates that RAD52 possesses multiple activities to cope with replication stress. For example, replication stress-induced DNA repair synthesis in mitosis (MiDAS) and oncogene overexpression-induced DNA replication are dependent on RAD52. RAD52 becomes essential in HR to repair DSBs containing secondary structures, which often arise at collapsed replication forks. RAD52 is also implicated in break-induced replication (BIR) and is found to inhibit excessive fork reversal at stalled replication forks. These various functions of RAD52 to deal with replication stress have been linked to the protection of genome stability at common fragile sites, which are often associated with the DNA breakpoints in cancer. Therefore, RAD52 has important recombination roles under special stress conditions in mammalian cells, and presents as a promising anti-cancer therapy target.

## 1. Introduction

Genome instability is a hallmark of cancer, and gross chromosomal rearrangements (GCR) are frequently found in the cancer genome [1,2]. DNA double-strand breaks (DSBs) are believed to be a major source of GCR, and faithful and accurate repair of DSBs is extremely important for the maintenance of genome stability and the prevention of cancer [3].

Homologous recombination (HR) emerges as a critical DSB repair pathway in mammalian cells [4]. Since HR uses sister chromatids as repair templates, it is thought to be the most accurate and conservative DSB repair pathway. One principal mechanism of HR is gene conversion (GC), which is utilized when both sides of a DSB contain homology to the donor [5,6,7] (Figure 1A). GC is initiated by 5’ to 3’ end resection, followed by invasion of the 3’ single-strand DNA (ssDNA) end into the donor to form a displacement loop (D-loop). The 3’ end of the invading strand is then used as a primer for new DNA synthesis. GC occurs mainly by synthesis-dependent strand annealing (SDSA) in mitotic cells, during which the other resected end anneals to the newly synthesized strand that has been displaced from its template. GC can also occur through the double-strand break repair (DSBR) pathway (also called the double Holliday junction pathway), wherein the double Holliday junction (dHJ) is formed by annealing of the displaced strand of the D-loop to the other resected end, which subsequently resolves to non-crossover (NCO) and crossover (CO) products. When only one DSB end has homology to the donor template, break-induced replication (BIR) is used (Figure 1B, [8,9,10,11]). The most specialized feature of BIR is DNA synthesis, where the primer presented at the invading strand is continuously extended. BIR is dependent on Pol32, the nonessential subunit of Polδ in yeast, and POLD3 in mammalian cells [12,13]. In contrast to GC, DNA synthesis in BIR often continues over a long distance and can proceed to the end of a chromosome in yeast [14,15,16]. 

The *RAD52* gene was initially identified in *Saccharomyces cerevisiae* in a genetic screen for mutants sensitive to ionizing radiation (IR) and was found to be critical for HR [17]. Based on a similar screen, a group of functionally connected genes that are important for HR were isolated and termed as the *RAD52* epistasis group, including *RAD50*, *RAD51*, and *RAD52* itself, *RAD54*, *RAD55*, *RAD57*, *RAD59*, *MRE11*, and *XRS2* [18]. Despite the critical role of yeast Rad52 in HR, it came as a surprise that RAD52 is dispensable for HR in mammalian cells. *RAD52* knockout mice are viable, exhibiting nearly normal phenotypes in DNA repair, HR and meiosis [19], which is in sharp contrast to the early embryonic lethality of *RAD51* knockout mice [20,21]. 

While mouse genetics indicates that *RAD52* is not an essential gene in mammalian cells, recent studies have led to new discoveries of RAD52 function in mammalian cells. The Powell lab found that RAD52 becomes essential for the viability of mammalian cells when BRCA1, BRCA2, and several other HR proteins are deficient [22,23]. The Hickson lab and the Halazonetis lab identified a role of RAD52 in promoting DNA synthesis following replication stress [24,25]. Our lab demonstrated that RAD52 has an indispensable role in repairing DSBs carrying DNA secondary structures at the ends with an implication for the protection of common fragile sites (CFSs) in the human genome [26]. These new findings have uncovered the biological functions of RAD52 in mammalian cells and suggest that RAD52 plays unique and important roles under special conditions such as replication stress. These studies also laid a foundation for establishing new anticancer therapies by targeting RAD52.

## 2. The Role of RAD52 in HR

RAD52 is composed of the N-terminal domain (NTD) and C-terminal domain (CTD) [27,28] (Figure 2A). While the RAD52 NTD is well conserved, with 42% identity shared by *S. cerevisiae* and humans, the CTD is very diverse. The NTD mediates DNA binding and is important for RAD52 multimerization to form a ring structure. Human RAD52 CTD is essential for its interaction with ssDNA-binding protein RPA [29] and RAD51 [30].

Biochemical studies show that *S. cerevisiae* Rad52 has two critical activities: mediator activity to promote the assembly of Rad51 filament on RPA-coated ssDNA and strand annealing activity to mediate annealing of two complementary ssDNA strands [31,32,33,34,35]. Although both *S. cerevisiae* and human RAD52 interact with RAD51 and RPA, the mediator activity was only observed in vitro for *S. cerevisiae* Rad52 but not for human RAD52 [33,34,35,36,37]. A study of the breast cancer suppressor BRCA2, a protein that is absent in yeast, revealed a central role of BRCA2 in HR [38], which likely replaces the mediator function of RAD52 for the loading of RAD51 onto RPA-coated ssDNA during HR in mammalian cells [36]. In contrast, strand annealing activity of RAD52 is well-conserved in different species [39]. In yeast and mammalian cells, RAD52 ssDNA annealing activity is responsible for single strand annealing (SSA) between repeated DNA sequences, a homology-mediated DSB repair pathway that is inhibited by RAD51 [40,41]. This ssDNA annealing activity is also needed for second end capture in DSBR and SDSA pathways in yeast [32,42]. Since HR is minimally affected by the loss of RAD52 in mammalian cells, RAD52’s role for second end capture also does not seem to be essential.

Although human RAD52 did not show recombination mediator activities in in vitro biochemical assays [36], depletion of RAD52 in BRCA1-, BRCA2-, and PALB2 (Partner and Localizer of BRCA2)-deficient cells results in decreased clonogenic survival, increased chromosomal abnormalities and reduced HR rates [22,23]. Meanwhile, damage-induced RAD51 nuclear foci are much reduced when RAD52 is depleted in BRCA2-deficient cells. These results suggest that RAD52 may serve as an alternative mediator pathway to BRCA2 to load RAD51 onto RPA-coated ssDNA. In chicken DT40 cells, RAD52 deficiency is synthetically lethal with inactivation of XRCC3 [43], suggesting that RAD52 may possibly act in conjunction with the RAD51 paralog complex RAD51B/C/D-XRCC2, given that XRCC3 exhibits similar synthetic lethality interactions with XRCC2 [44].

## 3. The Role of RAD52 in Break-Induced Replication (BIR) 

Despite the minor role of RAD52 in general HR in mammalian cells, growing evidence suggests that RAD52 is required for BIR. It has been proposed that BIR may promote DSB repair at single-ended DSBs that can arise at collapsed replication forks, or damaged telomeres that expose DSB ends [8,12]. Indeed, in mammalian cells, mitotic DNA synthesis (MiDAS), and alternative lengthening of telomeres (ALT) exhibit BIR features [25,45,46,47,48,49]. The Hickson group showed that MiDAS is activated by replication stress in mammalian cells and mainly occurs at “difficult-to-replicate” loci such as CFSs [25]. MiDAS is dependent on POLD3 and shows conservative replication features with DNA replication restricted to one chromatid. These characteristics suggest that MiDAS probably occurs through BIR. Replication stress may cause fork collapse at “difficult-to-replicate” loci, generating single-ended breaks which require repair via BIR at the early onset of mitosis. Interestingly, MiDAS is not dependent on RAD51 and BRCA2, but requires RAD52. Coincidently, ALT has also been shown to be independent of RAD51 but requires RAD52 [46]. 

In another study, the Halazonetis group showed that cyclin E overexpression induces DNA synthesis which is dependent on POLD3, RAD51 and BRCA2, suggesting the involvement of BIR for efficient DNA replication upon oncogenic stress [13]. Follow-up experiments demonstrated that RAD52 is required for cyclin E-induced replication and that RAD52 localizes to replication stress sites to promote replication restart [24]. They proposed that RAD52 functions together with POLD3 to facilitate BIR in mammalian cells and this RAD52-dependent BIR underlies the mechanism of oncogene-induced DNA synthesis.

## 4. The Role of RAD52 in the Protection of Structure-Prone DNA Sequences

CFSs often contain AT-rich sequences that are prone to forming DNA secondary structures [50]. These CFS-derived AT-rich sequences (CFS-ATs) stall DNA replication, leading to DSB formation [26,51,52,53]. By integrating these CFS-ATs into EGFP-based HR reporters, we found that CFS-ATs induce HR, which is further enhanced after hydroxyurea (HU) and aphidicolin (APH) treatment [26,51]. Oncogene overexpression also induces DSB formation and mitotic recombination at CFS-ATs. We previously showed that FANCM, a Fanconi anemia protein that possesses a translocase activity, is involved in fork reversal to remove DNA secondary structures formed at the replication forks, thereby protecting CFS-ATs [26]. When FANCM is deficient, DNA secondary structures at CFS-ATs are accumulated, leading to DSB formation and elevated mitotic recombination at CFS-ATs. Interestingly, we found that increased spontaneous and HU-induced- recombination at CFS-ATs due to a loss of FANCM activity is strongly suppressed by RAD52 inactivation [26]. Thus, although RAD52 is not needed for general HR in mammalian cells, it becomes indispensable for repairing DSBs generated at CFS-ATs. We further demonstrated that when DSBs contain secondary structures at the ends, RAD52 is required for HR even if the donor and recipient sequences contain perfect homologies. The working model is that when DNA secondary structures form at replication forks which could not be removed by FANCM, forks would stall and collapse, leading to the formation of DSBs that contain secondary structures at the ends, and these DSBs would require RAD52 to repair (Figure 2B). Therefore, FANCM and RAD52 play concerted roles in protecting CFS-ATs and other structure-prone DNA sequences. In support of this, we found that RAD52 is synthetically lethal with FANCM [26].

Mechanistically, it remains unknown why RAD52 is required for HR when the DSB ends contain secondary structures at the ends. It seems plausible that DNA secondary structures present at the ends may block efficient strand invasion and/or second end capture in HR, and under such conditions RAD52 is required (Figure 2C). We speculate that RAD52 may be needed for RAD51 to initiate strand invasion from a blocked end and/or to make the D-loop stable. RAD52 may use its ssDNA annealing activity to assist RAD51 with strand invasion and/or to stabilize the D-loop with an unpaired nonhomologous tail at the 3’ end. Alternatively, RAD52 may work together with RAD51 to initiate the pairing of the invading strand from the blocked end with the template strand by using its newly identified inverse strand exchange activity [54]. As revealed by single-molecule imaging, RAD52/RPA-ssDNA complexes are present as interspersed clusters in RAD51 filaments [55]. This supports the notion that RAD52 can function together with RAD51 for homology search and strand invasion, and this function of RAD52 may become essential when the DSB ends are blocked. As for second end capture in HR, RAD52 does not seem to be essential for repair of DSBs with perfect homology to the donor sequences, and other unknown mechanisms are likely involved in mammalian cells [56]. It is plausible that when DSB ends are blocked, the annealing activity of RAD52 may become indispensable for second end capture.

Rad1/Rad10 in yeast is responsible for cleavage of non-homologous tails at DSBs in SSA and GC [57,58,59]. This function is conserved in mammalian cells; the Rad1/Rad10 homologue ERCC1/XPF is also required for SSA and GC when DSB ends contain non-homologous tails [60,61,62,63]. We further demonstrated that ERCC1/XPF is involved in removing DNA secondary structures formed at DSB ends to facilitate HR [63]. It was described previously that RAD52 interacts directly with XPF and stimulates the endonuclease activity of ERCC1/XPF [64]. It remains possible that through the physical interaction, RAD52 facilitates ERCC1/XPF activity and, thus, is important for repairing DSBs with secondary structures at the ends. However, when we depleted RAD52 in *XPF* knockout cells, HR frequency was further decreased, suggesting that RAD52 has an activity non-overlapping with XPF (S. Li and X. Wu, unpublished). It is plausible that RAD52 carries functions in both promoting ERCC1/XPF activity to cut the secondary structures and facilitating RAD51-mediated strand invasion for HR after the blockage is removed.

The involvement of RAD52 in repairing DSBs with blocked ends may be linked to its function in BIR. As described in yeast, BIR is utilized in the situation when homology is present only at one DSB end or when single-ended DSBs are generated [8,9,10,11]. BIR is also used to repair DSBs with a large-sized (>1–2 kb) gap to the template (gap repair) [65,66]. Conceivably, when non-homologous tails or secondary structures are present at DSB ends, one DSB end is blocked and cannot be recognized, and consequently a BIR-like mechanism may be engaged. This hypothesis will need to be tested in a defined reporter model system. 

## 5. The Role of RAD52 in Coping with Replication Stress 

Accumulating evidence points to an important role of RAD52 in coping with replication stress. RAD52 is required for replication stress-induced MiDAS and is also important for induced DNA synthesis in response to oncogenic stress due to cyclin E overexpression [24,25]. BIR is proposed to be used in MiDAS and oncogene-induced DNA synthesis and thus RAD52-dependent BIR is especially important under replication stress. Our findings that RAD52 is required for repair of DSBs with DNA secondary structures at ends suggest another important function of RAD52 in dealing with replication stress [26]. CFS-ATs are prone to forming DNA secondary structures, which cause replication stalling and fork collapse especially under replication and oncogenic stress. Repair of DSBs at these structure-prone DNA sequences requires RAD52. Additionally, CFS-ATs, structure-prone DNA sequences, are prevalent in our genome. For example, among DSB sites that have been mapped genome-wide following replication stress, about half (>30,000 sites) contain AT-rich sequences that are present at both CFSs and previously defined early replication fragile sites (ERFSs) [67]. Meanwhile, more than 700,000 sequences are predicted to form G-quadruplexes (G4s) in the human genome [68]. Since replication forks would stall at these sites upon replication stress, we anticipate that the role of RAD52 in repairing DSBs generated upon fork collapse at these sites constitutes an important part of RAD52 activity to cope with replication stress. Recently, a new study has revealed one more role of RAD52 in protecting genome stability upon replication stress [69]. RAD52 binds to the stalled replication forks and limits the loading of SMARCL1 (SWI/SNF Related, Matrix Associated, Actin Dependent Regulator of Chromatin, Subfamily A-Like 1), also known as HARP to prevent excessive fork reversal activity. This RAD52 function is important in protecting replication forks from unscheduled degradation. Collectively, RAD52 possesses multiple activities at replication forks to preserve fork stability in response to replication stress.

## 6. The Connection of RAD52 with CFS Protection

CFSs are large chromosomal regions that are susceptible to forming gaps and breaks upon mild replication stress [70]. They are part of normal chromosome structures and are present in all human individuals. However, during the early stage of cancer development, they become unstable prior to other places in the genome and are often associated with chromosomal rearrangement sites in tumors [71,72,73]. It is believed that CFS instability is one driving force for tumorigenesis. Different models have been proposed to account for the underlying mechanisms of CFS instability. Late replication timing, a paucity of replication origins, replication stalling by AT-rich sequences and collision between replication and transcription all contribute to fragility of CFSs [52,74,75,76,77,78]. 

Compelling evidence suggests that perturbation of DNA replication at CFS-ATs is an important cause of CFS instability. It has been shown that CFS-ATs placed on plasmids in yeast cause replication stalling [53]. Single-molecule analysis using SMARD also demonstrated that in human cells, fork arrest at the FRA16C site is preferentially close to the AT-rich sequences [52]. We further showed that DSBs caused by fork stalling at CFS-ATs are repaired by HR which is dependent on RAD52 [26]. Inactivation of RAD52 therefore results in DSB accumulation at CFS-ATs. These data support the model that fork collapse at CFS-ATs in S-phase which is mediated by MUS81 endonuclease cleavage does not require RAD52, but subsequent repair does involve RAD52 [26,63]. In addition to its role in repair of DSBs containing secondary structures, the role of RAD52 in BIR would also be important for replication restart at CFSs. Along this line, loss of RAD52 would cause accumulation of DSBs or chromosome breakages at CFSs in S-phase.

MiDAS is mainly observed at CFSs when cells are under replication stress in S-phase [45]. Replication at CFSs is often late and replication stress further delays it. It is proposed that underreplicated DNA is cleaved by MUS81 in mitotic prophase, followed by BIR to complete DNA replication and repair, which is shown as MiDAS. Intriguingly, MUS81 recruitment to CFSs depends on RAD52, and in RAD52-deficient cells, underreplicated DNA at CFSs is not cleaved by MUS81 [25]. Consequently, RAD52 depletion causes increased frequency of ultra-fine DNA bridges (UFBs) at CFSs and chromatin bridges in anaphase due to incompletion of DNA replication at CFSs. In addition to MUS81, ERCC1/XPF is also suggested to cleave underreplicated DNA regions at CFSs [79]. Once underreplicated regions of CFSs are cleaved, RAD52 function in BIR is also expected to be important for the completion of DNA replication and DSB repair at CFSs in early mitosis. 

Collectively, the role of RAD52 in the protection of genome stability at CFSs is multifaceted (Figure 3). The repair function of RAD52 in HR at structure-prone DNA sequences and its role in BIR are expected to prevent chromosome breakage at CFSs, whereas its role in MUS81 recruitment in early mitosis to cleave underreplicated DNA at CFSs would minimize chromosome mis-segregation and non-disjunction.

## 7. RAD52 as a New Target for Cancer Therapy

Replication stress is highly associated with tumorigenesis [80,81]. Oncogene activation disturbs DNA replication and causes replication stress in multiple ways, such as by causing premature origin activation, impaired replication progression and accumulation of reactive oxygen species that damage replication forks [81,82,83,84]. Since RAD52 is involved in protecting stalled replication forks and repairing collapsed forks, cancer cells may rely more on RAD52 than normal cells to survive due to oncogenic replication stress. In this respect, we found that accumulated DSBs at CFS-ATs caused by oncogene overexpression depend on RAD52 for repair [26]. It has also been shown that RAD52 is needed for cyclin E-overexpressing cells to progress through G1 to S phase and *RAD52* knockout reduces tumor progression in *APC* (adenomatous polyposis coli gene) mutant mouse tumor models [24]. Given that RAD52 is a non-essential gene for normal cells but is indispensable for tumor cells to survive, RAD52 presents as a good target for cancer treatment. 

Previous studies have shown that RAD52 is required for viability of BRCA1- and BRCA2-deficient cells [22,23], suggesting a targeted treatment opportunity via inhibition of RAD52 to specifically kill cancer cells with BRCA deficiency. This provides another therapeutic strategy alternative to PARP inhibitors to treat BRCA-deficient tumors. In our study, we identified a synthetic lethal interaction between RAD52 and FANCM, suggesting that inactivation of RAD52 can also eradicate FANCM-deficient cells while leaving normal cells unaffected. Like the BRCA proteins, FANCM is a breast cancer susceptibility gene, and its deficiency has been found in breast tumors, especially triple-negative breast cancer [85,86,87]. In addition, FANCM deficiency has been described in high grade serous ovarian cancer and sporadic head and neck squamous cell carcinoma [88,89]. Inhibition of RAD52 can thus be used as a new targeted therapeutic strategy for the treatment of FANCM-deficient tumors, which is expected to have low toxicity to normal cells.

Substantial efforts have been made to identify inhibitors for RAD52 inactivation. A small peptide has been designed and shown to disrupt the assembly of the RAD52 ring structure [90]. Small molecule inhibitors have also been screened by different research groups for disruption of RAD52 binding to ssDNA and inactivation of ssDNA annealing of RAD52. A number of RAD52 inhibitor leads were reported and showed effects to selectively kill BRCA1- and BRCA2-deficient cancer cells [91,92,93,94]. However, the RAD52 inhibitors identified so far are not potent enough to be applied to clinical usage.

## 8. Conclusions and Perspectives

The role of RAD52 in mammalian cells has been puzzling for years due to the viability of RAD52 knock-out mice and the lack of strong DNA repair phenotype in RAD52-deficient cells. The findings of the synthetic lethal interactions between BRCA1/2 and RAD52 suggest a backup activity of RAD52 in HR in mammalian cells. Recent discoveries of RAD52 involvement in BIR, in repairing DSBs containing secondary structures at their ends and in protecting stalled replication forks have further revealed critical roles of RAD52 in preserving replication fork integrity and in coping with replication stress. The key remaining questions are how RAD52 is involved in BIR and in repairing DSBs blocked by DNA secondary structures mechanistically, and whether the stand annealing activity of RAD52 is needed for these functions. Since RAD52 is not essential for normal cell viability, but becomes indispensable under stress conditions or upon loss of function of other key repair proteins such as BRCA1, BRCA2, and FANCM that are associated with cancer, RAD52 is a promising therapeutic target in cancer treatment. Given that RAD52 lacks enzymatic active sites, it is still challenging to identify potent small molecule inhibitors for RAD52.

## Figures and Tables

**Figure 1 cancers-11-01467-f001:**
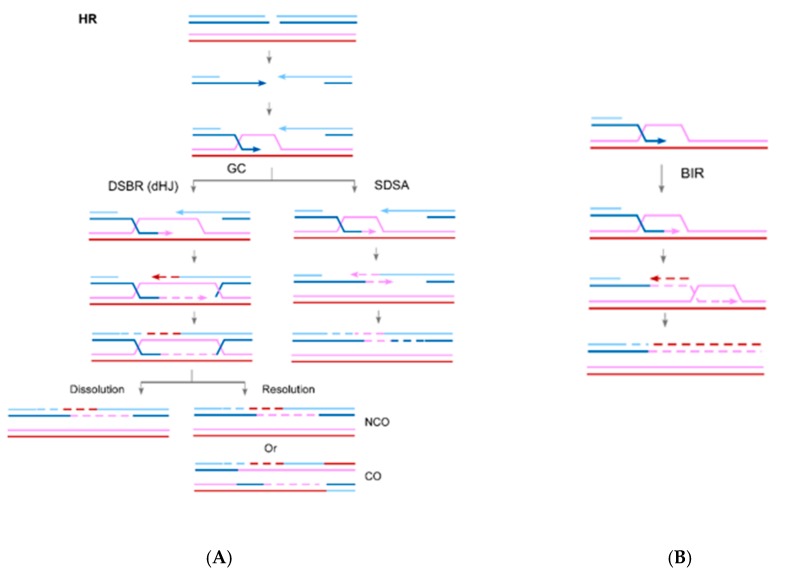
Pathways of homologous recombination (HR). Gene conversion (GC) is used when both sides of a DSB contain homology to the donor (**A**). GC can occur in the forms of synthesis-dependent strand annealing (SDSA) or double-strand break repair (DSBR) (also called the double Holliday junction repair: dHJ repair). Double Holliday junctions can be resolved by dissolution to give non-crossover (NCO) products, or by resolution to produce NCO and crossover (CO) products. When only one DSB end has homology to the donor template, break-induced replication (BIR) is used (**B**).

**Figure 2 cancers-11-01467-f002:**
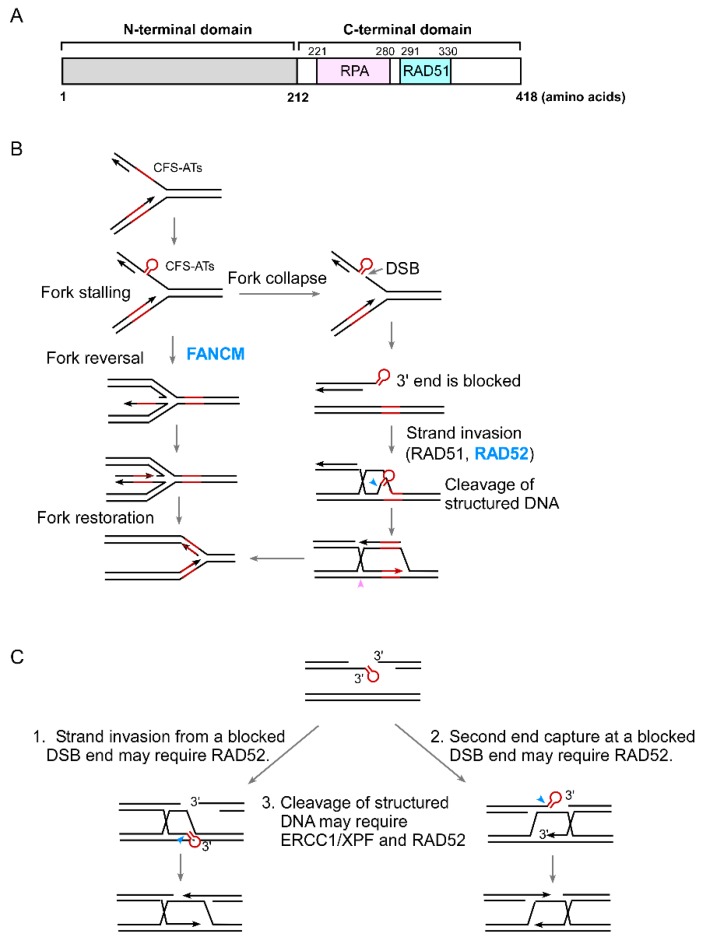
Proposed models for the role of RAD52 in repair of DSBs containing DNA secondary structures. (**A**) The domain structure of human RAD52. RPA and RAD51: domains of RAD52 that interact with RPA and RAD51, respectively. (**B**) The model of the roles of FANCM in removing DNA secondary structures on the forks and RAD52 in repairing DSBs with secondary structures at the ends. (**C**) Proposed mechanisms for how RAD52 is involved in repair of DSBs carrying DNA secondary structures.

**Figure 3 cancers-11-01467-f003:**
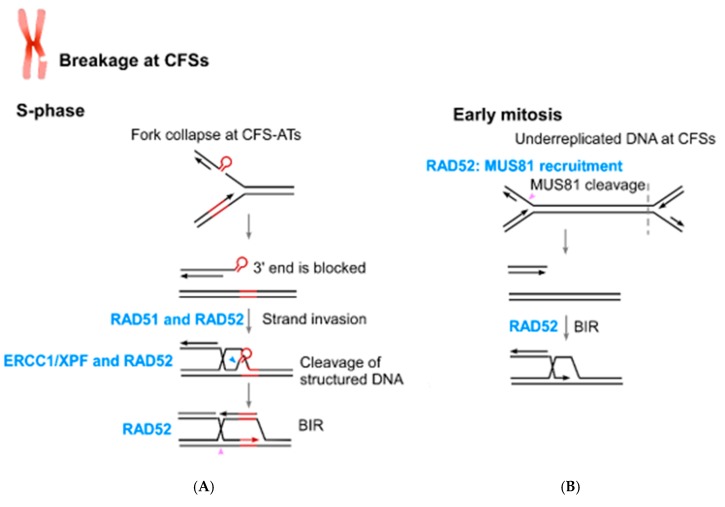
Multiple functions of RAD52 in coping with replication stress contribute to the maintenance of CFS stability. RAD52 is required for repairing DSBs at CFS-ATs and promoting BIR upon fork collapse at CFSs in S-phase (**A**). RAD52 is also required for recruiting MUS81 to cleave underreplicated DNA at CFSs and promote BIR in early mitosis (**B**).

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
