# Peer review of "Replication Stress Response Links RAD52 to Protecting Common Fragile Sites"

_cancers, 2019, doi:10.3390/cancers11101467_

Round 1

Reviewer 1 Report

In this review, t
he authors summarize the roles of Rad52 in replication stress . Ove rall,
the description is clear and precise. The citations are fair and appropriate. But in some
parts, their models and arguments are unclear and raise critical questions. These points
should be addressed and well explained within the review
Figure 2B:
How does
seco ndary structure d ssDNA lead to DSB s during replication? Why do the
problems on the lagging strand, not leading strand, stall the replication machinery? Why
can’ t replication associated helicases resolve this kind of DNA structure problems? The
replication machinery can go through this problem only to create some gaps on lagging
strand, which can be easily repaired by repair systems other than HR.
Is this
Rad52 f unction upstream or downstream of Rad51?
Why is secondary structured ssDNA cleaved by nucleases rather than resolved by
helicases? It can be deleterious.
Line 161:
The data described here suggest that RAD52 is functioning only after FANCM
depletion. Can all CFS ATs be repaired by FANCM pathway without RAD52 in normal
cells? Does this mean RAD52 is just a part of the backup pathway?
L
ine 179 It is not clear why strand exchange activity can explain the role of RAD52 in
strand invasion with seco ndary stru ctured ssDNA.
Figure 3
left : Indicating t he site of MUS81 cleavage would be helpful.
Line 295: Is there any evidence for this argument, reliance of cancers on RAD52? If yes,
that should be cited.
Line 333:
Again, most of the roles described here is in the backup pathways. Therefore,
it could be overstatement to say these roles of RAD52 in preserving replication fork
integrity and in coping with replication stress are critical.
L
ine 341 : It is challenging, b ut a couple of recent studies suggest that RAD52 is
targetable by some small molecules. The authors should include these findings in the
discussion.

Author Response

Reviewer #1:

Figure 2B:

How does secondary structured ssDNA lead to DSBs during replication? Why do the problems on the lagging strand, not leading strand, stall the replication machinery? Why can’t replication-associated helicases resolve this kind of DNA structure problems?The replication machinery can go through this problem only to create some gaps on lagging strand, which can be easily repaired by repair systems other than HR.

Response: During DNA replication, ssDNA is exposed mainly at the lagging strand. When DNA is in the ssDNA form, AT-rich sequences (or other usual DNA sequences) would have chance to form DNA secondary structures (when they are not in the DNA duplex form). We added this description in the text (L156). Apparently, DNA replication helicases are not sufficient to resolve secondary structures at these unusual DNA sequences during replication. We found that FANCM is required to resolve DNA secondary structures at CFS-ATs, while previous studies showed that helicases PIF1 and FANCJ are involved in resolving G4 secondary structures.  

Is this Rad52 function upstream or downstream of Rad51?

Response: In Figure 2C, we showed three possible mechanisms by which RAD52 promotes HR at DSBs containing secondary structures. The role of RAD52 in facilitating strand invasion or stabilizing D-loop is at the same stage as RAD51, whereas the role of RAD52 in second end capture or cleavage of secondary structures is downstream of RAD51. The description is in the text (L199 and L205).

Why is secondary structured ssDNA cleaved by nucleases rather than resolved by helicases? It can be deleterious.

Response: Secondary structures are only cleaved by nucleases when helicases/translocases such as FANCM are deficient or when cells are under replication stress, which may overwhelm the helicases/translocases-involved pathways.

Line 161:The data described here suggest that RAD52 is functioning only after FANCM depletion. Can all CFS-ATs be repaired by FANCM pathway without RAD52in normal cells? Does this mean RAD52 is just a part of the backup pathway?

Response: In the paragraph starting on L154, we mentioned that DSBs can be generated spontaneously at CFS-ATs, which would be enhanced after HU or APH treatment or oncogene expression. Loss of FANCM would also increase DSB formation at CFS-ATs. Since RAD52 is essential to repair DSBs carrying secondary structures at the ends, RAD52 is required for repairing DSBs at CFS-ATs once they are generated, which can be induced by replication stress, loss of FANCM or other conditions. Thus, it is not true that RAD52 functions only after FANCM depletion; RAD52 is also critical for repairing DSBs at CFS-ATs upon replication stress such as HU and APH treatment or oncogene expression.

Line 179:It is not clear why strand exchange activity can explain the role of RAD52 in strand invasion with secondary structured ssDNA.

Response: We propose that RAD52 uses its ssDNA annealing activity to assist RAD51 for strand invasion and/or stabilize the D-loop when unpaired nonhomologous tails at the 3’ end make the D-loop unstable.

Figure 3left: Indicating the site of MUS81 cleavage would be helpful.

Response: We indicated that in the figure.

Line 295: Is there any evidence for this argument, reliance of cancers on RAD52? If yes, that should be cited.

Response: RAD52 is needed for cyclin E-overexpressing cells to progress through G1 to S phase and RAD52 knockout reduces tumor progression in APC mutant mouse tumor models (L312-315).

Line 333: Again, most of the roles described here is in the backup pathways. Therefore, it could be overstatement to say these roles of RAD52 in preserving replication fork integrity and in coping with replication stress are critical.

Response: As described above, RAD52 is essential to repair DSBs formed at DNA secondary structures including CFS-ATs especially upon HU or APH treatment (replication stress) or oncogene expression (oncogenic stress). Thus, RAD52 is not only involved in backup pathways, but has indispensable roles under the condition of replication stress.

Line 341: It is challenging, but a couple of recent studies suggest that RAD52 is targetable by some small molecules. The authors should include these findings in the discussion.

Response: Small molecule inhibitors are mentioned in references 91-94.

Reviewer 2 Report

Although roles of Rad52 in yeast recombination are established, its exact role in higher eukaryotes remains unclear. However, several lines of recent evidence suggest the mechanisms underlying Rad52-dependent homologous recombination in mammals. Among these, this review focuses on its roles in break-induced replication. In the introduction section, the functional difference between yeast and human Rad52 is described. Next, replication stress-induced mitotic DNA synthesis and cyclin E overexpression-induced replication are introduced. Then, how Rad52 is involved in break-induced replication and the protection of common fragile sites is explained. Finally, the therapeutic strategy targeting Rad52 is introduced. This review is very informative.

Minor points

Line 12: “in mitotic” is “in mitosis”. Line 46: “Pol in yeast” is “Pol delta in yeast”. Line 98: A space between Rad and 51 should be deleted.

Author Response

Reviewer #2:

Thank the reviewer for the careful editing and I have made corrections for all the points raised.

Reviewer 3 Report

Dear Dr Wu,

I read your review on Rad52 with great interest. The key points that are the novel roles of Rad52 in the repair of replication-induced DNA lesions and the processing of terminally blocked DNA breaks possibly through the recruitment of nucleases are well described.

I have no scientific comments as the review covers the current knowledge as well as the work by others comprehensively. I like that the names of the other research groups are included.

My comments relate only to some text editing and to making the content more accessible for readers who are less familiar with the field.

Comments:

L10: BRCA (BReast CAncer genes)

L13: replication are dependent

L15 BIR (Break Induced Replication)

L17: the DNA breakpoints

L19: presents a promising anti-cancer target.

L30: delete "typically"

L33: contain homology to the intact region on the sister chromatid (donor)

L39: been returned from its repair template

L42: of the D-loop

L44: is used where the primer presented by the invading strand is continously extended

L47: often contious over a long

L52: to the donor (intact sisterchromatid) (left)

L58: in a genetic screen

L60: on a similar screen, g group of functionally connected genes

L68: functions

L69: The Powell group

L71 The Hickson group and the Halazonetis group identified a role

L79: therapies by targeting

L83: While the RAD52 NTD

L84: humans, the CTD is is very diverse.

L85: The NTD ... to form a ring complex.

L86: RPA (DNA single-strand binding protein)

L97: a mediator activity .... Rad51 filaments

L101: A study of the breast cancer

L103: for the loading of RAD51 onto

L111: to be essential or second end capture is not an important repair pathway in mammalian cells.

L113: in in vivo

L114: please introduce PALB2 to the reader

L118: to BRCA2 to load RAD51 onto RPA-coated single-stranded DNA

L129: The Hickson group

L1343: MiDAS occurs probably through BIR.

L139: Halazonetis group

L144: functions

L149: Common Fragile Sites (CFSs)

L153: after replication arrest induced by hydroxyurea (HU) and replication instability triggered by aphidicolin (APH)

L155: protein that posesses

L166: structures form at replication forks, which

L203: activity to cut the secondary structure ... for HR after the blockage was removed.

L221: of DSBs with terminal DNA secondary structures suggest another

L232: replication blockage

L236: please introduce SMARCL1 to the reader

L242: Common Fragile Sites (CFSs) are

L244: human genomes.

L245: unstable earlier than other regiions in the genome

L258: results therefore in DSB

L259: S-phase is mediated by MUS81 endonuclease cleavage

L270: cells underreplicated DNA

L301: please introduce APC to the reader

L310: cells while leaving normal cells unaffected.

L310: Like the BRCA proteins, FANCM is a

L311: in tumours, especially triple-negative breast cancers

L315: a new therapeutic strategy for the treatment of FANCM-defective tumors

L320: different research groups

L332: structures at their ends

L340: is a promising therapeutic target

Author Response

Reviewer #3:

Thank the reviewer for the careful editing and I have made corrections for most points. Please see below for explanations of several points.

L33: contain homology to the intact region on the sister chromatid (donor)

L52: to the donor (intact sisterchromatid) (left)

Response: For gene conversion, I agree with the reviewer that the sister chromatid is often used as the template. However, sister chromatids are not used in all cases and homologous sequences in the genome can also be used as templates. Since we describe gene conversion as a general term here to cover all situations, I would not emphasize the use of sister chromatid as a donor in this context.

L149: Common Fragile Sites (CFSs)

L242: Common Fragile Sites (CFSs) are

Response: CFSs are defined on line L137.

Reviewer 4 Report

The author presents a comprehensive review of RAD52 function in homology-dependent repair and the replication stress response. The review is timely given the current interest in RAD52 as a target for anti-cancer therapy. I have a few minor suggestions/corrections:

Figure 2: Several other helicases are implicated in fork reversal. Do any of the others exhibit genetic interaction with RAD52?

Line 109: Lao et al. (2008) Mol Cell provided in vivo support for RAD52 function in second end capture.

Line 259: Change This to These (data is plural)

Author Response

Referee #4:

Figure 2: Several other helicases are implicated in fork reversal. Do any of the others exhibit genetic interaction with RAD52?

Response: We are still analyzing whether other helicases such as SMARCAL1 and ZRANB3 in removing DNA structures by fork reversal and their genetic interactions with RAD52. More thorough analyses would be needed before we can discuss the details in the review.

Line 109: Lao et al. (2008) Mol Cell provided in vivo support for RAD52 function in second end capture.

Response: Thank the reviewer for suggesting this reference and we have cited their work in the review.

Line 259: Change This to These (data is plural)

Response: Thank the reviewer for the careful editing and I have made the correction.